# Quadrivalent Vaccines for the Immunization of Adults against Influenza: A Systematic Review of Randomized Controlled Trials

**DOI:** 10.3390/ijerph19159425

**Published:** 2022-08-01

**Authors:** Alice Mannocci, Andrea Pellacchia, Rossella Millevolte, Manuela Chiavarini, Chiara de Waure

**Affiliations:** 1Faculty of Economics, Universitas Mercatorum, 00186 Rome, Italy; alice.mannocci@unimercatorum.it; 2Department of Medicine and Surgery, University of Perugia, 06132 Perugia, Italy; rossella.millevolte@studenti.unipg.it (R.M.); manuela.chiavarini@unipg.it (M.C.); chiara.dewaure@unipg.it (C.d.W.)

**Keywords:** influenza vaccination, quadrivalent influenza vaccines, adults, immunogenicity, systematic review

## Abstract

Vaccination is the most effective intervention to prevent influenza. Adults at risk of complications are among the targets of the vaccination campaigns and can be vaccinated with different types of quadrivalent influenza vaccines (QIVs). In the light of assessing the relative immunogenicity and efficacy of different QIVs, a systematic review was performed. Randomized controlled trials conducted in adults aged 18–64 years until 30 March 2021 were searched through three databases (Medline, Cochrane Library and Scopus). Twenty-four RCTs were eventually included. After data extraction, a network meta-analysis was not applicable due to the lack of common comparators. However, in the presence of at least two studies, single meta-analyses were performed to evaluate immunogenicity and efficacy; on the contrary, data from single studies were considered. Seroconversion rate for H1N1 was higher for standard QIVs, while for the remaining strains it was higher for low-dose adjuvanted QIVs. For seroprotection rate, the recombinant vaccine recorded the highest values for H3N2, while for the other strains, the cell-based QIVs achieved better results. In general, standard and cell-based QIVs showed an overall good immunogenicity profile. Nevertheless, as a relative comparative analysis was not possible, further research would be deserved.

## 1. Introduction

Influenza is a major cause of disease burden globally [1]. Worldwide, annual epidemics are estimated to result in about 3 to 5 million cases of severe illness, and about 290,000 to 650,000 respiratory deaths [2]. The estimated mean annual influenza-associated respiratory excess mortality rate ranges at 0.1–6.4 per 100,000 individuals for people younger than 65 years, 2.9–44.0 per 100,000 individuals for people aged between 65 and 74 years, and 17.9–223.5 per 100,000 for people older than 75 years [3].

Two influenza A subtypes (A/H1N1 and A/H3N2), and two antigenically and genetically distinct B lineages (B/Victoria and B/Yamagata) are the current seasonal influenza strains circulating since 1985. Both influenza A subtypes and both B lineages co-circulate, and their frequency distribution varies widely by season and geographic region [4].

Influenza vaccination is the most effective intervention available to prevent influenza infection and its complications and represents a major public health initiative [2].

The Advisory Committee on Immunization Practices (ACIP) recommends influenza vaccination to all people aged 6 months and older, without any contraindications [5].

As influenza viruses rapidly mutate, and the prophylactic effects of vaccination wane over time, vaccine formulations are updated each year in compliance with the World Health Organization (WHO) and Committee for Medicinal Products for Human Use (CHMP) recommendations (in the EU) [2,5].

Up to 2012, influenza vaccines have been produced to protect against three different seasonal influenza viruses, namely A/H3N2, pandemic A/H1N1, and one out of the two influenza B lineages [6].

The first quadrivalent influenza vaccines (QIVs) were licensed in the US in 2012 [7,8]. Currently, in most high-income countries, quadrivalent vaccines are used, which protect against influenza A/H3N2, pandemic A/H1N1 and both influenza B lineages. The immunogenicity and efficacy of available vaccines varies by age, individual immune response, and the degree of cross-protection against alternate lineages in the case of trivalent vaccines [9,10,11]. Therefore, for each recipient, a licensed and age-appropriate vaccine should be used.

Even if it is challenging to compare the immunogenicity and efficacy of different types of influenza vaccines [12], there are already reviews on this issue in pregnant women, children < 5 years old [13], and the elderly [14].

Together with children aged 6 months–5 years, pregnant women, and the elderly, the adult population also represents a target group of influenza vaccination [5].

In adulthood, inactivated QIVs have already been compared with inactivated trivalent influenza vaccines (TIVs) with respect to immunogenicity and safety [15]. Nevertheless, because influenza vaccines are expected to be quadrivalent and there are different types of QIVs that have been licensed or are under study [5], we proposed summarizing the evidence on their immunogenicity and efficacy in adults aged 18–64 years and comparing them through a systematic review and a network meta-analysis (NMA). The expected aim is to provide useful evidence to identify QIVs that are more efficacious in eliciting an immune response and avoiding influenza in adulthood.

## 2. Materials and Methods

### 2.1. Protocol Registration

The systematic review was conducted in accordance with the Preferred Reporting Items for Systematic Review and Meta-Analyses 2020 (PRISMA 2020) guidelines [16] and registered in the PROSPERO database (number: CRD42021243135).

### 2.2. Search Strategy and Data Source

A comprehensive literature search was carried out from databases’ inception up to 30th March 2021. Medline, Scopus and Cochrane Central Register of Controlled Trials databases were used to identify all the original articles investigating the immunogenicity or the efficacy of any QIV in people aged from 18 to 64 years. The medical subject headings (MeSH) and key words used for the search are reported in the Appendix A. The search was limited to human studies and studies published in the English language.

### 2.3. Studies Selection

Two authors (AP, RM) screened the articles on the basis of titles and abstracts first, using Rayyan. Any double publication of a study was recorded and thereafter removed. Full texts of potential eligible papers were then obtained and checked for final inclusion according to the inclusion and exclusion criteria. Articles were included if they met the following criteria:Evaluated any QIV compared to placebo or TIV or another QIV in adults aged 18–64 years;Adopted a randomized controlled trial (RCT) study design that compared two or more groups, one of which receiving any QIV as intervention;Issued results on immunogenicity, in terms of seroconversion (SCR) or seroprotection rate (SPR), and efficacy, in terms of laboratory-confirmed influenza.

The SCR was defined as the percentage of participants with either a pre-vaccination HI titer <10 and a post-vaccination HI titer ≥40, or a pre-vaccination HI titer ≥10 and a 4-fold increase in hemagglutination inhibition (HI) antibodies titer after vaccination. The SPR was defined as the percentage of participants with a HI titer ≥40. If the same study was reported in several publications, we selected the most recent publication. Reviews and meta-analyses were excluded. Disagreements were resolved by discussion or in consultation with a third author (CdW or MC).

### 2.4. Data Extraction and Quality Assessment

For each included study, two authors (AP, RM) independently carried out data extraction and quality assessment. From each included study, the following information was extracted: first author’s last name, year of publication, trial status, country, study population, population characteristics (gender, age), participants in the experimental and control arms (number and characteristics), type of intervention, type of control, study endpoints with definition, and results for each endpoint stratified by influenza strains (A/H1N1, A/H3N2, B/Victoria and B/Yamagata). Data on immunogenicity and efficacy, referring only to adult populations, were extracted from per protocol populations. For studies with missing or incomplete information, authors were contacted. If attempts to contact authors were not successful, such studies were excluded from the quantitative analyses. If raw data were not available, they were calculated from the percentages reported in the study.

The study quality was assessed through the Cochrane Risk of Bias Tool (RoB-Tool 2) [17] by two authors (AP, RM) independently; any disagreement was solved by discussion and consultation with a third author (CdW or MC). Studies were assessed for potential sources of bias according to five key domains, namely, the randomization process, deviations from intended interventions, missing outcome data, measurement of the outcome, and selection of the reported results. A judgment of low risk, some concerns, or high risk was released for each key domain. The overall risk of bias for each included study was considered to be low if all domains were judged to be at low risk, some concern if the study was judged to raise some concerns in at least one domain, but not to be at high risk of bias for any domain, or high if it was judged to be at high risk of bias in at least one domain or it was judged to have some concerns for multiple domains in a way that substantially lowered confidence in the results. Studies were not excluded because of quality issues.

### 2.5. Statistical Analysis

The NMA was carried out using the Markov Chain Monte Carlo engine WinBUGS free software if the assumption of transitivity was satisfied [18]. In the absence of this assumption, a meta-analysis of the following endpoints was performed, considering only data from the arm receiving any QIV as intervention: SCR, SPR and efficacy (laboratory confirmed influenza).

Results were reported as percentage with 95% confidence intervals (95% CI); the width and potential overlaps of 95%CI were considered to elaborate on the comparisons among QIVs.

Meta-analyses were performed using StatsDirect statistics software version 3, and a fixed- or random-effect model was used to pool data based on the assessment of heterogeneity. Heterogeneity was measured using Cochran-Q test and inconsistency index (I^2^) method [19,20]. An inconsistency index with a value >90% was considered an indicator of substantial heterogeneity among studies, preventing the pooling of data [21]. Low heterogeneity was considered if I^2^ was <25%, and the fixed-effect model was then applied. In case of moderate (25–50%) and high heterogeneity (>50%), the random-effect model was used.

## 3. Results

### 3.1. Studies Selection

From the literature search through Medline (n = 518), Scopus (n = 1090) and Cochrane (n = 757) databases, and after removing duplicates (n = 930), we identified 1435 records for title and abstract screening (Figure 1). Among these, 1400 articles were excluded because they were reviews, case studies, case reports or commentaries, or they did not investigate the populations and outcomes of interest. Thirty-five articles were subjected to full-text review. Subsequently, 11 articles were excluded because of the following reasons: six showed overlapping data, three did not have available full texts and two did not present relevant results. Therefore, at the end of the selection process, 24 RCTs [22,23,24,25,26,27,28,29,30,31,32,33,34,35,36,37,38,39,40,41,42,43,44,45] met the inclusion criteria and were eventually included (Figure 1).

### 3.2. Characteristics of Included RCTs

Amongst the included studies, all RCTs have been completed. Eleven studies (45.8%) were published between 2011 and 2016 [22,23,24,25,26,27,28,29,30,31,32], and thirteen (54.2%) after 2016 [33,34,35,36,37,38,39,40,41,42,43,44,45]. Seven studies (26.9%) were conducted in multiple countries [25,26,28,29,39,41,45]. Most of the studies were conducted in the USA (11, 45.8%) [22,24,25,27,28,30,31,34,35,36,45], five (19.2%) in Germany [25,26,39,41,45], four (15.4%) in South Korea [25,33,38,43] and in Canada [28,32,42,45], and two (7.7%) in Finland [44,45], the Philippines [29,45], Belgium [39,41] and France [26,39]. Twenty-three studies (95.8%) were conducted in both women and men, with one (4.2%) in women only [44]. All studies were conducted in adult subjects; among them, nine studies (47.5%) were conducted in people ≥18 years of age [24,25,26,28,31,36,39,40,41], and five (20.8%) were on the 18–49 age group [22,27,32,34,42]. All data were extracted from adult population groups, except for two studies [25,28] that reported data only for a ≥18-year-old population, therefore including older people.

As far as the intervention was concerned, 14 studies (58.3%) assessed the standard-dose egg-based QIVs [23,24,25,26,28,29,36,37,38,39,40,41,43,44], 3 (12.5%) the plant-derived QIVs [32,42,45], 2 (8.3%) the cell-based QIVs [31,33], 2 (8.3%) the live attenuated QIVs [22,27], 2 (8.3%) recombinant QIVs [34,35], 1 (4.2%) the low-dose adjuvanted QIV (assessed together with the standard-dose egg-based QIV in the same study) [23], 1 one (4.2%) the intradermal QIV [30].

The control arm received the analogous TIV except for two studies on recombinant QIVs that compared it with the standard-dose egg-based QIVs [34,35], and the three studies on the plant-derived QIVs that had placebo in the control arm [32,42,45].

All the studies evaluated the SCR. The serological outcome was determined by HI assay at 21–28 days after vaccination. The SPR was evaluated in most studies, excluding Dunkle et al. [34] (recombinant QIV in 18–49 age group) and Block et al. [22] (live attenuated QIV). Furthermore, in one RCT on live attenuated QIV, it was defined in a different way with respect to the other studies where the HI titer cut-off was set at 32 instead of 40 [27].

The efficacy endpoint, namely laboratory-confirmed influenza, was assessed using real-time reverse transcriptase polymerase chain reaction (RT-PCR) and reported by only two studies [35,45]. The characteristics of the included RCTs are reported in Appendix A.

### 3.3. Quality Assessment of Included RCTs

Nineteen studies (79.2%) had a low risk of bias [23,24,25,26,27,28,29,30,31,33,34,35,36,37,39,40,41,43,45], whereas five studies (20.8%) raised some concerns [22,32,38,42,44], four of which were due to the randomization process [22,32,38,44] and one for missing outcome data [32] (Figure 2).

### 3.4. Synthesis of Data

The assumption of transitivity was not satisfied and the NMA was not applicable.

As far as the meta-analysis of immunogenicity and efficacy endpoint was concerned, 14 studies on the standard-dose egg-based QIV, and 2 on the cell-based QIV were considered for both SCR and SPR. For the latter, two studies on the live attenuated QIV were also considered for meta-analysis. For other QIVs, data extracted from single studies were reported (Table 1 and Table 2). The combination of data on laboratory-confirmed influenza was not possible, because only two studies assessed the efficacy to prevent laboratory-confirmed influenza: one on recombinant QIV [35] and one on plant-derived QIV [45].

The SCR for strains A/H1N1 ranged from 5% for live attenuated QIVs to 65% for standard-dose egg-based QIVs, whereas values for A/H3N2 ranged from 5% for live attenuated QIVs to 66% for low-dose adjuvanted QIVs; for B/Yamagata strain, SCR ranged from 9% for live attenuated QIVs to 79% for low-dose adjuvanted QIVs; eventually, for B/Victoria strain, SCR ranged from 10% for live attenuated QIVs to 65% for low-dose adjuvanted QIVs. Looking at the 95%CI, it can be observed that standard-dose egg-based, low-dose adjuvanted, cell-based, recombinant and intradermal QIVs showed similar SCRs with respect to both A strains, apart from cell-based QIVs with respect to H3N2 that showed a lower immunogenicity. With respect to B lineages, results are less conclusive, but it appears that overall low dose adjuvanted QIVs showed a better profile with respect to B Yamagata.

Regarding SPR, values ranged from 25% to 98% for A/H1N1, respectively, for live attenuated and cell-based QIVs, and from 26% to 100% for A/H3N2, respectively, for live attenuated and recombinant QIVs. As for B lineage, the lowest values of SPR were observed for recombinant QIV and plant-derived QIV, namely 68% for B/Yamagata and 41% for B/Victoria. The highest values were attained by the cell-based QIVs (99% for B/Yamagata and 98% for B/Victoria). Looking at the 95%CIs, cell-based and recombinant QIVs showed similar SPR with respect to A/H1N1. Low-dose adjuvanted QIVs added to them regarding A/H3N2. With respect to B lineages, low-dose adjuvanted and cell-based QIVs showed similar results. Additionally, the standard-dose egg-based QIVs showed similar results as compared to them with respect to B/Yamagata.

With respect to laboratory-confirmed influenza, for plant-derived QIVs, the attack rate in the vaccine group was 4.5% (215/4812), and in the placebo group it was 7.3% (251/4812), with an absolute vaccine efficacy of 38.8% (95%CI 17.9–48.7) to prevent influenza caused by any strain and of 35.1% (95% CI 17.9–48.7) to prevent influenza caused by vaccine-matched strain. Regarding the recombinant QIV, in the 50–64-year-old subgroup, the attack rate was 1.7% for the recombinant QIV group, and 2.9% for the standard QIV control group, leading to a relative vaccine efficacy of 42% (95% CI 15–61).

## 4. Discussion

Influenza continues to be a major public health problem worldwide, and current influenza vaccines remain a valuable public health tool to counter it.

In this regard, the CDC estimated that influenza vaccination averted 60,500 deaths in the United States between the 2010–2011 and 2019–2020 seasons [46].

In Europe, it is estimated that seasonal influenza vaccination prevents an annual average of 1.6–2.1 million cases of influenza, 45,300 to 65,600 hospitalizations, and 25,200 to 37,200 deaths [47].

QIVs have currently replaced TIVs in several countries, but national immunization advisory technical groups do not generally make specific recommendations for the preferential use of one over other. Nevertheless, the assessment of their relative immunogenicity and efficacy could be useful to inform decisions.

Our systematic review and meta-analysis collated together immunogenicity and efficacy data of different QIVs. The analysis of data issued by included RCTs basically showed that QIVs have different immunogenicity. In particular, differences can be observed among QIVs in terms of SCR and SPR against the same vaccine strain, but also within the same QIV, with respect to different endpoints and vaccine strains.

In general, live attenuated QIVs did not show good immunogenicity, whereas standard-dose egg-based QIVs showed better SCR, and cell-based QIVs showed better SPR for all four strains. In addition, the recombinant QIV induced high SPR for A/H3N2. Actually, the development of vaccines based on recombinant protein and viruses grown on cell lines has been pursued to overcome the problem of mismatch between vaccine and circulating strains, which mostly regards H3N2 strains [48] and is due to the phenomenon of egg adaptation.

It is known that SCR and SPR are considered by regulatory agencies to decide upon market authorization of influenza vaccines. In fact, the relationship between HI antibody titer and clinical protection has been well-established. In healthy adults, a serum HI titer ≥40 is associated with a >50% reduction in influenza infection or disease and is considered as a surrogate correlate of protection [49,50]. This association is also supported by a modelling study, finding that, regardless of the vaccination status of individuals and the viral strains (either A or B), a positive and significant relationship exists between HI antibody titer and clinical protection against influenza infection [51].

According to the results of our systematic review, and considering that an HI titer ≥40 is considered the best available parameter for predicting protection from influenza infection, it can be said that the cell-based QIVs showed the best immunogenicity profile even though other QIVs, with the exception of the live attenuated, also achieved high SPR with some differences across strains. However, considerations should be given to the criticalities of the evaluation of immunogenicity, such as the low sensitivity to B strains and the high degree of interlaboratory variability [52]. Furthermore, new-generation influenza vaccines call for a deeper understanding of immune responses and further evaluation of correlates of protection [50]. This has been showcased by the live attenuated influenza vaccines that were proved to provide high protection despite low HI seroresponse rate [53].

As a matter of fact, since 2017, new CHMP guidelines recommended the measurement of neutralizing antibodies in addition to the HI titer and encouraged the assessment of broader immune responses through anti-neuraminidase antibodies, antibody kinetics, and cell-mediated immunity [54]. This is of utmost importance in light of the use of alternative vaccine manufacturing platforms, such as plants that can allow a large-scale production of recombinant proteins, thus also being important in the case of the need for a pandemic vaccine or vaccine shortages [55]. Additionally, adjuvants and alternative routes of administration, namely intradermal ones, could allow for antigen dose-sparing effects, thus being promising approaches for large-scale production [56,57]. Continuous updates of the evidence on these new generation vaccines is indeed relevant.

Beyond the assessment of immunogenicity, efficacy and effectiveness data on laboratory-confirmed influenza and influenza-related clinical endpoints could further provide evidence for the relative comparison of QIVs. In this regard, a recent retrospective study assessed the relative effectiveness of cell-based QIVs, compared to the standard egg-based QIVs, among people aged 4–64 years in the United States during the 2019–2020 influenza season. Results showed that cell-based QIVs were significantly more effective than standard-dose egg-based QIVs in preventing influenza-related and respiratory-related hospitalizations/emergency room visits [58].

In discussing the results of our systematic review, several limitations should be taken in consideration. The most important is represented by the potential heterogeneity among studies due to different influenza seasons and different participants’ history before vaccination. Another relevant aspect is that all the included studies were conducted in the Northern Hemisphere, and this might limit the generalizability of results. Furthermore, even though HI antibody responses were commonly assessed at 21 days post-vaccination, in a few studies, they were assessed at 28–30 days post-vaccination.

Among the strengths of our systematic review, the following could be included: the search was performed on the most relevant databases, and the selection of studies and data extraction was performed by taking appropriate measures to prevent potential errors. Therefore, selection bias can be ruled out. Furthermore, included studies were mostly at a low risk of bias, and this enables us to consider their results robust.

## 5. Conclusions

To the best of our knowledge, this is the first study that collates immunogenicity data of the different QIVs that are either in use or under study for the immunization of the adult population. Although a relative comparison among QIVs was not possible, the assessment of SCR and SPR and their 95%CI provides an overview of their respective potentiality in adulthood.

## Figures and Tables

**Figure 1 ijerph-19-09425-f001:**
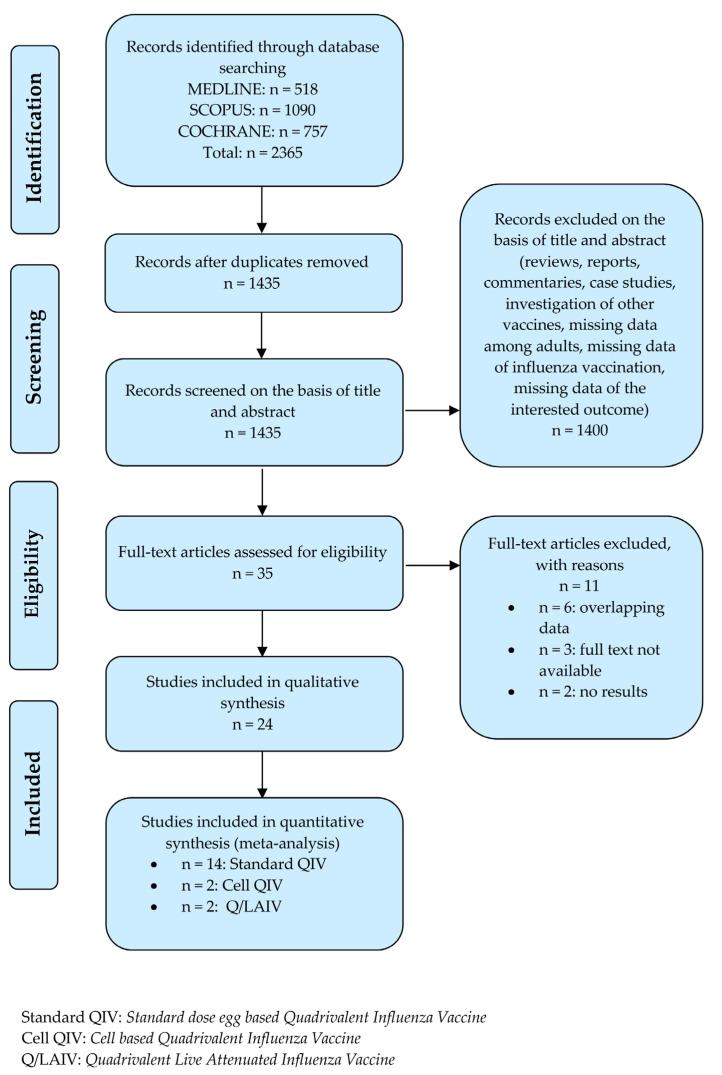
Flow diagram of the systematic literature search.

**Figure 2 ijerph-19-09425-f002:**
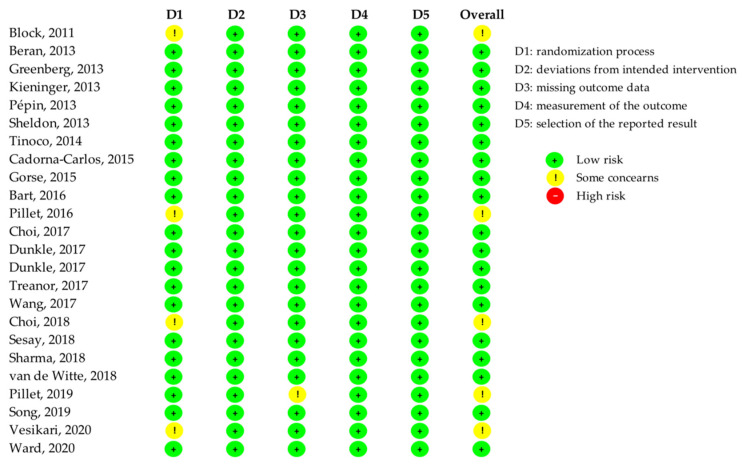
Risk of bias of included studies.

**Table 1 ijerph-19-09425-t001:** Type of vaccine and seroconversion rate (SCR) for the four strains A/H1N1, A/H3N2, B/Yamagata and B/Victoria {Number of studies}.

Type of Vaccine [ref]	SCR (95%CI) {n}			
	H1N1	H3N2	B Yamagata	B Victoria
Standard QIV [23,24,25,26,28,29,36,37,38,39,40,41,43,44]	65% (58–72%) {14} *	65% (58–72%) {14} *	63% (58–68%) {14} *	63% (59–67%) {14} *
Low Dose Adjuvanted QIV [23]	58% (48–67%) {1}	66% (57–75%) {1}	79% (70–86%) {1}	65% (56–74%) {1}
Cell based QIV [31,33]	58% (47–68%) {2} *	51% (47–56%) {2} *	48% (40–57%) {2} *	53% (50–56%) {2} *
Plant-derived QIV [45]	37% (32–43%) {1}	46% (40–52%) {1}	32% (26–37%) {1}	18% (14–23%) {1}
Recombinant QIV [35]	56% (49–63%) {1}	63% (56–69%) {1}	43% (36–50%) {1}	26% (20–33%) {1}
Intradermal QIV [30]	58% (55–61%) {1}	59% (56–61%) {1}	56% (53–59%) {1}	50% (47–53%) {1}
Live attenuated QIV [22,27]	5% (4–6%) {2} °	5% (4–6%) {2} °	9% (7–11%) {2} °	10% (6–15%) {2} *

SCR was defined as the percentage of those with either a prevaccination HAI titer <10 and a post vaccination HI titer ≥40, or a prevaccination HAI titer of ≥10 and a ≥ 4-fold increase in HI titer after vaccination. * pooled proportion using random effects: Cochran Q *p* < 0.01 and/or I^2^ > 50%; ° pooled proportion using fixed effects.

**Table 2 ijerph-19-09425-t002:** Type of vaccine and seroprotection rate (SPR) for the four strains A/H1N1, A/H3N2, B/Yamagata and B/Victoria {Number of studies}.

Type of Vaccine	SPR (95%CI) {n}			
	H1N1	H3N2	B Yamagata	B Victoria
Standard QIV [23,24,25,26,28,29,36,37,38,39,40,41,43,44]	65% (58–72%) {14} *	94% (88–98%) {14} *	96% (93–99%) {14} *	81% (68–91%) {14} *
Low Dose Adjuvanted QIV [23]	88% (81–94%) {1}	98% (94–100%) {1}	95% (90–98%) {1}	97% (92–99%) {1}
Cell based QIV [31,33]	98% (97–99%) {2} °	99% (98–99%) {2} °	99% (98–99%) {2} °	98% (93–100%) {2} *
Plant-derived QIV [45]	75% (69–80%) {1}	85% (81–89%) {1}	76% (71–91%) {1}	41% (35–47%) {1}
Recombinant QIV [35]	91% (88–94%) {1}	100% (99–100%) {1}	68% (63–73%) {1}	50% (44–56%) {1}
Intradermal QIV	N.A.	N.A.	N.A.	N.A.
Live attenuated QIV [27]	25% (23–28%) {1}	26% (23–28%) {1}	79% (76–81%) {1}	56% (53–58%) {1}

SPR was defined as the percentage of participants with a HI titer of ≥40 (except for Sheldon et al. [27] on live attenuated QIVs with a HI titer of ≥32). N.A.: not available; * pooled proportion using random effects: Cochran Q *p* < 0.001 and I^2^ > 50%; ° pooled proportion using fixed effects.

## Data Availability

Not applicable.

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
