# Peer review of "Quadrivalent Vaccines for the Immunization of Adults against Influenza: A Systematic Review of Randomized Controlled Trials"

_ijerph, 2022, doi:10.3390/ijerph19159425_

Round 1
Reviewer 1 Report
I have one question/comment.
Why did authors present common data of seroconversions for both groups in the Table 1: the first one - for people with prevaccination HAI titre <10 and the other one - >10 with 4-fold increase in HA titer after vaccination. I think that it will be interesting to present data separately.
Reviewer 2 Report
In their systematic review, Mannocci et al. compiled all available data on relative immunogenicity and efficacy of different quadrivalent influenza vaccines (QIVs) for adult population. The study was conducted in accordance with the PRISMA 2020 guidelines and included only high-quality research. The authors present the immunogenicity data (seroconversion rates and seroprotection rates) and, where possible, efficacy data (laboratory confirmed influenza). This data set enables an assessment of which vaccine type is preferable in the adult population. Overall, the manuscript is well written and contains all the necessary descriptions that formed the basis of the authors' conclusions. There are only minor comments that can be considered for further improvement of the quality of the paper:
1. It seems important to include a start date for searching articles, not just an end dateÑŽ
2. The quantitative immunogenicity analysis was performed for three vaccine types (standard QIV, cell-based QIV and Q/LAIV), however the abstract refers to a low-dose adjuvanted QIV. Please ensure that there are no confusions between main text and the abstract.
3. Lane 92. In terms of…
4. Lanes 93, 130. When using the terms “laboratory-confirmed influenza” it is important to specify that this is an efficacy endpoint.
5. Lane 134. “usingStatsDirect” a space is missing.
6. Lane 293. “at 95%Cis,” – please correct.
